# Last Aid Course—The Slovenian Experience

**DOI:** 10.3390/healthcare10071154

**Published:** 2022-06-21

**Authors:** Erika Zelko, Larisa Vrbek, Melita Koletnik

**Affiliations:** 1Institute for General practice, Johannes Keppler University, 4020 Linz, Austria; 2Department for Family Medicine, Faculty of Medicine University Maribor, Taborska cesta 8, 2000 Maribor, Slovenia; 3Cerebral Paralysis Association of Slovenia, Rožanska ulica 2, 1000 Ljubljana, Slovenia; vrbek.larisa@gmail.com; 4Faculty of Arts, University of Maribor, 2000 Maribor, Slovenia; melitakk@gmail.com

**Keywords:** palliative care, education, lay public, Last Aid course

## Abstract

Educating and raising awareness among lay members of the public about palliative care can significantly improve the care for terminally ill patients and their quality of life. This paper reports on the survey aimed at assessing the experience and expectations of participants in the Last Aid course launched in Slovenia in 2019 to train hospice volunteers and promote dialogue on death and dying. The course implementation was supported by materials prepared, translated, and/or adapted from German under the PO-LAST project, which linked Slovenian medical and healthcare professionals, hospice representatives, and university students. The Last Aid course follows an international four-module curriculum that has been successfully applied in 18 countries so far. In Slovenia, the course was delivered 30 times with 21 in-person deliveries and 9 online events attended by 450 participants of different sexes, ages, and professions. The surveyed population included 250 people who returned the evaluation questionnaires by October 2020. The aim of the analysis was to gain insight that can be applied broadly in future work and research on adult education on palliative care and the erasure of death-related taboos.

## 1. Introduction [

Globally, life expectancy has been steadily increasing over the past two centuries. Based on the latest United Nations Population Division Estimates [1] the average life expectancy at birth in 2022 amounted to 73.2 for both sexes and was the highest in developed eastern and western countries, e.g., Hong Kong with 85.3 or Switzerland with 84.7 years. Driven by increases in life expectancy and by falling fertility rates, the world population is aging at an accelerated pace. If, a few decades ago, the most common cause of death in the developed world was infectious disease, today, it is noncommunicable illnesses, such as heart disease, stroke, cancer, and diabetes, which according to the WHO data [2], collectively accounted for more than 70% of all deaths worldwide—even during the recent COVID-19 pandemic. 

In modern society, death is an important process. However, talking about death is still painful and agonizing for most people [3] and commonly referred to by negative metaphors, e.g., “a foe to be conquered” or the “loss of the battle”, frequently resulting from the long “war against disease” [4]. Moreover, in our consumer society, death is often marginalized [5] and placed outside the confines of our everyday existence.

A similar, uneasy status is attributed to palliative care, which is regularly conceptualized as specialized (medical) care that supports the patients diagnosed with life-threatening and terminal illnesses, provides relief from pain and other symptoms, and aims to improve the quality of life for both patients and their families “by means of early identification and impeccable assessment and treatment of pain and other problems” [6]. Palliative care is a holistic approach that actively addresses an individual’s total needs—physical, psychological, social, and spiritual suffering of patients and psychological, social, and spiritual suffering of family members—and should be based on good teamwork and collaboration between its providers [7]. Among the general public, there is substantial misunderstanding of and lack of awareness about palliative care [8], and lay people are frequently unaware of its fundamental principles. Consequently, palliative care is underused [9,10]. A terminal or life-threatening illness is an illness or condition which cannot be cured and is likely to lead to someone’s death [9]. End-of-life care is defined as support for people who are in the last months or years of their life [11,12]. Last Aid comprises both palliative and end-of-life care, and offers support to patients who need long-term care or palliation to the very end of their lives, in life situations where death is known to be close, and in the dying process.

Greater public awareness, education, and training, on the other hand, support the implementation of palliative care. These elements positively affect people’s quality of life and contribute to wider acceptance of death as a natural part of life [13]. 

The idea of a Last Aid course and a public knowledge approach to palliative care were first described by the future emergency care physician and consultant Georg Bollig in the late 2000s. The original aim was to discontinue the discourse on death and dying as a taboo within many communities and to teach the public about palliative care. Consequently, the course educates participants about palliative and end-of-life care, while at the same time, it provides information on how and where to obtain help from professionals and the local community. The Last Aid course is modelled on the First Aid course because Last Aid should be considered as important as first aid, with which we all have been acquainted at some point in our personal or working lives [14].

On the other hand, we know little about dying and death, although at some time, we all will be faced with the process and the transience of our loved ones or ourselves. End-of-life care, when partnered with community efforts to provide support and care for death, dying, and loss, leads to the establishment of compassionate communities, as described by Kellehear [15], whose members can participate in the provision of palliative care within the limits of their skills and abilities [16].

The recent COVID-19 pandemic exposed the vulnerability of the individual, e.g., to mental health problems [17,18], and of the society, e.g., to highly disrupted rehabilitation services, including the delivery of specialist palliative care [19]. Developing truly compassionate communities which can face illness, death, and bereavement as part of the life cycle and preserve human dignity to the very end helps build resilience on the personal and the societal level, with all members contributing to the best of their ability to the good of all.

## 2. Barriers to Palliative Care, Compassionate Communities, and the Last Aid Course

The barriers to accessing palliative care are very well-known, and the policy reports describing them keep emerging. Very often, relatives who need to care for a loved one diagnosed with a life-threatening and/or terminal illness face a crisis. Dionne-Odom et al. [10], e.g., reported that more than one-half of unpaid family caregivers in the US caring for relative with a medical, behavioral, disability, or other condition had never heard of palliative care. Even among those who had, most had not distinguished it from hospice care and death. Based on their findings, authors appealed for more awareness of palliative care among the lay public, focusing on the needs of the patient and their family. Further to this, Bollig [16], acknowledged an urgent need to educate nonprofessionals in palliative care and end-of-life care. The involvement of lay people in palliative care is necessary and important for several reasons, which range from demographic changes, increasing numbers of elderly suffering from chronic diseases, and a shortage of care professionals, to changes in family structure and the preservation of human dignity of the dying.

Nevertheless, raising public awareness on such a difficult subject requires a subtle approach. Compassionate communities, pioneered by A. Kellehear in the mid-2000s, are perfectly positioned to achieve this by placing palliative care within everyone’s responsibility [15]. Compassionate communities are defined as naturally occurring networks of support in neighborhoods and communities, surrounding those experiencing death, dying, caregiving, loss, and bereavement [20]. They are formed around a group of people who are concerned about the quality of life of the community’s members and play a much stronger role in the care of both people at end-of-life and their families. However, compassionate communities represent only one of the four pillars that comprise palliative and end-of-life care. The others are specialist palliative care, i.e., a medical specialty within medical and nursing training programs; generalist palliative care, i.e., care at the end-of-life that is not provided by specialist palliative care teams; and civic end-of-life care offered or rather supported by, e.g., schools, workplaces, or churches. Their effective coordination contributes significantly to access to palliative care and the wellbeing of the terminally ill [20]. 

In their recent article, reimagining access to palliative care, Abel, Kelleher, Mills, and Patel added that “the community itself must become part of the palliative care offer/provision to help regulate and mediate its own internal differences and services”. This should go beyond merely volunteering toward recognizing that every citizen has a role to play in palliative care, thus allowing that community engagement evolves into “community development and self-provision (changing the power dynamics)” of end-of-life care [21]. 

An important achievement supporting the objective that palliative care become everyone’s responsibility is the Last Aid course, a recent course concept for educating the public about palliative care. It is based on the premise that knowledge about palliative care should become part of public education, which is frequently very limited or totally absent in most communities [16], and relies almost exclusively on personal experience as the main source of information [22]. Stemming from the idea presented by Bollig, the course encourages public discourse on taboo topics of death and dying and educates the lay public about them [13,16,23].

The method pursued by the Last Aid courses is a combination of situated learning, which is conditioned by interaction between participants and builds on prior knowledge and experience [24], and of community of practice, where participants with similar interests are brought together to improve their knowledge for practice and future action [25].

The course follows a public knowledge approach and consists of four modules: Care at the End of Life, Advance Care Planning and Decision Making, Symptom Management, and Cultural Aspects of Death and Bereavement. Comparable to the Chain of Survival in emergency medicine, the Chain of Palliative Care is used to visualize the networks of palliative care in a community. The Chain, presenting the cooperation between nonprofessionals and healthcare professionals in palliative care, establishes a link to Kellehear’s pillars that make up palliative care, and the public knowledge approach centered on widespread education of the lay public aims to change attitudes and behaviors in the community in the direction of a more positive attitude toward participating in palliative care.

It is important that the course be adapted to the cultural diversity and constraints of the environment. Thus far, localized versions of Last Aid courses have been implemented and well-received in several countries including, e.g., Australia, Ireland, Brazil, Switzerland, Scotland, Germany, and Denmark. Furthermore, 99% of Last Aid course participants say they would recommend them to others [26].According to them, the course contributes to improving the discussion about death and dying in families, at the workplace and elsewhere. The participants’ overall response has been thus very positive [27]. A special edition of the Last Aid course was recently developed to address bereaved children and teenagers. A pilot study showed it was met with a positive response and proved that most children and teenagers want to talk about death and dying [28].In total thus far, Last Aid courses have been attended by over 26,000 people, and more than 2000 Last Aid trainers have been educated to implement Last Aid activities in their local communities [16].

## 3. Organization of Palliative Care in Slovenia, Last Aid Slovenia, and the PO-LAST Project

In Slovenia, the strategy for palliative care is set out in the National Program for Palliative Care [12].The plan requires all hospitals to provide from 10 to 15 beds for palliative patients, depending on the demographic and socio-economic profile of the region where the hospital is located. Consequently, the two main medical centers in Ljubljana and Maribor have two separate palliative care wards, while smaller hospitals provide individual palliative care beds. Additionally, the hospice house *Hiša Ljubhospica* was built in 2010 in Ljubljana, and six mobile palliative care teams attend to palliative care cases throughout the country. Nevertheless, most patients in need of palliative care are cared for by palliative care nurses and family doctors. Some are placed in nursing homes or have home caregivers. No official data on how many patients receive palliative care in Slovenia are available, but it is estimated that some 20% of those in need of palliative care also receive it [29].

The first edition of the Last Aid course was organized in 2019 by the Institute for Palliative Medicine and Hospice Care at the University of Maribor’s Faculty of Medicine Maribor. Since then, it has been delivered, in cooperation with the Slovenian Lions clubs and the Slovenian Hospice Association, twenty-five times over a period of two years, with five deliveries taking place online during the COVID-19 pandemic. A special online course (webinar) was developed for the deaf and hard of hearing using sign language so that an even more diverse population could be reached. To allow for unimpeded interaction and active participation, the number of participants did not exceed 15 per delivery. The professional profiles of the participants included, e.g., medical and healthcare students and practitioners (nurses and physicians), teachers, social workers, business professionals, lawyers, engineers, priests and nuns, homemakers, and people without current employment.

The course implementation was supported by materials prepared, translated, and/or adapted from German under the Slovenian government-financed Assistance for a Better Quality of Life in the Last Phase of Life (PO-LAST) project, which connected Slovenian medical and healthcare professionals, hospice representatives, and university teachers and students. The Slovenian Hospice Association and University of Maribor’s Medical Faculty and the Faculty of Arts cooperated with the project, as well as Alma Mater Europaea, an independent higher learning institution.

The main objective of the project was to provide information and/or access to information for people with life-threatening illness and their caregivers, which would also be of use for the implementation of the Last Aid course. The thirteen project members (1) prepared the Slovenian Atlas of Palliative Care [30], dealing with support options and information sources for the terminally ill and their caregivers; (2) translated the manual for Last Aid instructors from German into Slovene and adapted the content for the Slovenian context; (3) prepared a brochure with essential information on palliative care in Slovenia; and (4) organized and implemented dissemination workshops. The four-month project was successfully concluded in June 2020. All materials that were produced during the project support the implementation of Last Aid courses in Slovenia, which have been, thus far, received extremely well by the participants, as is detailed below.

## 4. Methods

To gain a more in-depth insight into the implementation of the Last Aid course in Slovenia, we decided to analyze the participants’ responses in 2021. Our overarching research question was to investigate their experience and to identify suggestions for improvement of the course. For this purpose, we used the pre-prepared international Last Aid questionnaire with open-ended and closed-ended questions. The used tool was a validated questionnaire including both quantitative and qualitative data, which has been previously used in Germany, Switzerland, and Austria [27]. The questionnaire was administered to our participants immediately after the course, irrespective of whether it had been held online or in-person, and it was sent or given to them with documents confirming their participation on the course. In total, 386 questionnaires were distributed personally or sent out by e-mail, of which 250 were returned. The response rate was thus 64.7%. The non-respondents were not analyzed, and we cannot speculate or draw any conclusions about the reasons why the questionnaires were not returned. All participants were informed that participation in this evaluation was voluntary and that they could choose not to complete the questionnaire if they did not want to. Data were collected from June 2019 to December 2021.The questionnaire was divided into three sections. The first section included four closed-ended questions evaluating the total course as well as individual modules on a 5-point rating scale. The second section summarized personal reflections of the participants and consisted of two open-ended questions and four yes or no questions exploring the participants’ understanding of the specific topics covered in the course, while an additional open-ended question elicited the “highlights” of the course and suggestions for future work. The third section collected the participants’ demographic data. Our sample was limited with the available resources. Thus, we are aware that the participants of the course and of our study do not constitute a representative sample of the entire Slovenian population and cannot be immediately generalizable to the entire population. However, we believe the number of respondents to be large enough to give the data validity and attention. The advertising for the course was interest-based (e.g., to hospice volunteers and healthcare providers), and we used chain referral. Registration and attendance were voluntary, and the size of individual course groups was limited to 15 participants as recommended by Last Aid International.

We used descriptive statistics combined with qualitative analysis of the open-ended questions to investigate the data obtained. Included in the analysis were 250 questionnaires: 243 were fully answered and 7 were missing one answer, but we nevertheless considered these relevant and included the answers in the analysis of the relevant sections. The coding of qualitative answers was carried out by three independent researchers. For quantitative answers, statistical significance was assumed at the *p* < 0.05 level. Descriptive answers to open-ended questions were qualitatively thematically analyzed to identify, analyze, and report themes, i.e., patterns, within the data. A cumulative understanding of the results was created by identifying themes and patterns and organizing them into categories. We believe that the methodological conditions (qualitative and quantitative) guarantee the overall quality of the study in terms of internal validity (risk of bias), external validity (generalizability), and reporting quality. The quantitative and qualitative results are mutually supportive, and it is our belief that they will contribute to the further improvement of the Last Aid course and deliver suggestions for advancement to the Last Aid International group, which is responsible for the international course development and quality.

## 5. Results

Table 1 below shows basic course participants statistics.

Table 2 presents participant ratings for the entire course and for the individual modules. In relative terms, most participants rated both the entire course (87.7%) and individual modules (more than 75% for all modules) as “excellent” (5).

Based on the age of the participants, we defined three age brackets: young (under 35), middle-aged (35–64), and elderly (65+). We used these to compare the participants’ evaluations of the overall course and the individual modules to establish any age-based differences. Table 3 presents the ratings by age bracket. The results showed that satisfaction both with the course as a whole and with individual modules was highest among middle-aged participants, who on average assessed the course better than the young or the elderly.

On average, female participants rated the entire course better than their male counterparts (4.89 vs. 4.69). The average grades for individual modules also differed by sex and its represented in Table 4.

Given the very limited number of participants with primary education (only three), we subsequently created only two categories to analyze education-based differences in satisfaction with the course: secondary or lower education, and higher education. The average scores showed that better educated participants rated the total course more favorably than those with less education (4.86 vs. 4.83). However, these differences were small and not statistically significant. The results were similar with regard to the individual modules. On average, the more educated participants rated the modules higher than the participants with a high school degree or lower. The most significant differences in participants’ average scores by education category were found in relation to Module 2 (Advance Care Planning and Decision Making) (0.12), and the least significant in relation to Module 3 (Symptom Management) (0.03). The difference for Module 1 (Care at the End of Life) amounted to 0.06 and for Module 4 (Cultural Aspects of Death and Bereavement) to 0.07.

A total of 185 (72.5%) participants answered the open-ended question about which topic particularly spoke to them, what they missed from the course, and their opinion about it. The results are represented in Table 5.

After conducting a detailed thematic analysis, the following two themes were identified: time, with participants’ comments related to different periods of taking care of patients diagnosed with a life-threatening and/or terminal illness, and suggestions for improvement of the course in Slovenia. The responses were grouped into the following categories: (1) period before death; (2) period after death; and (3) support for the bereaved and perceived needs as the disease progresses. The second theme, suggestions, included the following categories: ethical and cultural aspects, regular education, spirituality, and children. Almost a quarter of participants (23.1%) suggested that such courses should be held more frequently or on a regular basis to raise awareness and educate people. The following two salient statements testify to this: “…the topic should be taught in secondary schools, e.g., in the school of nursing…” (No. 135) and “A lecture that makes you realize there are more important things in life because too often we don’t realize that death is part of life, and we don’t learn enough about it at the faculty” (No. 101). Additionally, the participants recognized the benefits, quality, and usefulness of the course, with over 99% of them writing that they would recommend it to their friends. Other interesting ideas and suggestions were advanced by the participants in regard to the future implementation of the course. “There is too little talk about the subject; when my relative was dying in hospital, I didn’t know where to turn for information, all [I got] referred to his medical condition. To spread the news about the course in the community” (No. 123). Other participants suggested an “upgrade and expansion of the module” (No. 10) on how to “communicate with aggressive relatives, and more [information] on spiritual care” (No. 54). Finally, combined quantitative and qualitative results showed that the Last Aid course was useful for the participants, with almost 95% of them confirming that they had heard and acquired new information and knowledge.

## 6. Discussion

Palliative and hospice care education initiatives, such as the Last Aid course, are needed to increase awareness of and reduce misperceptions about palliative and end-of-life care conceptions and services reported by more researcher [8,31,32,33]. The introduction of the Last Aid course in Slovenia, supported by materials prepared by the PO-LAST project, has aimed to achieve just this. 

In terms of general population statistics, statistically significantly more women attended the course (Table 1), which was expected. In Slovenia, women are more likely to be employed as health and midwifery care professionals and health and midwifery care assistants. According to the data of the Statistical Office of the Republic of Slovenia, at the end of 2019, “87.5% of nursing and midwifery professionals were women and only 12.5% were men, while 82.5% of associate professionals were women and 17.5% were men” [34]. In average all participants rated the course and the modules very positive (Table 2). Statistically significantly more higher-educated participants and those aged between 35 and 64 were more satisfied with the course (Table 3 and Table 4). According to Slovenian statistic agency, 24.5% of Slovenians are higher educated. Adapting the course in an easy language may help to reach less educated people. In particular, the modules Care at the End of Life and Advance Care Planning and Decision-making showed a slight positive deviation from satisfaction with other course modules as reported by participants. Further research is needed to account for this difference. The structure of the participants and their responses are not surprising in relation to international research, which shows that also informal caregivers are most often middle-aged women [35,36,37].

As presented above, the evaluation of the course by the participants showed a high level of satisfaction with both the content and the delivery of the course. The average rating of the total course was “excellent” (5) as evidenced by 87.7% of the returned questionnaires. Moreover, 75% of the questionnaires also rated individual modules as “excellent” (Table 2). Socializing and sharing of personal experience on the course were seen as a major benefit of the course by our participants. They also gave interesting suggestions and orientations for future work. Similar results have been reported by researchers in other countries, e.g., Germany, Denmark, and Switzerland, as described by [26]. Furthermore, in line with results from other countries, the Slovenian participants advised expanding some of the currently addressed topics and organizing courses in schools [28]. Based on this, we can assume that in Slovenia, there is growing recognition of the need to “normalize” death and to provide opportunities for individuals and communities to recognize death and dying as a social process, in line with observations made by Abel and Kellehear [38]. The need to educate the public about palliative and hospice care and to increase public awareness of and reduce misconceptions about the topic were also reflected in our participants’ qualitative answers, which included, e.g., the appeal to “spread the news about the course in the community”. Their opinion corresponds to the recent findings by Zelko and her colleagues in the Slovenian context that, in order to widely implement palliative care in Slovenia, the awareness of and education about palliative care need to be improved [39]. Consequently, if we want to successfully raise awareness of this topic, which is still too frequently taboo, we need to find ways to reach out to the lay public and involve them in the process of exchanging views and information. “Cultural interventions” such as Death Cafés [40], and intensified promotion of palliative care by healthcare providers and education about death and dying, as well as the creation of truly compassionate communities [41], are all examples of strategies that aim to jointly encourage individuals and communities to reflect on death and respond to end-of-life issues. Another promising example of good practice is the Japanese educational program using stories as a primary learning tool, which showed that it is possible to deepen understanding among the lay public of the concept of end-of-life care through a narrative [9]. It is important that palliative care education initiatives be adapted to the cultural context of the environment where they are delivered, as highlighted by, e.g., Shen et al. [42], Isaacson [43], Hayes et al. [11], or in the Australian context by McGrath and Holewa [44]. The particulars for such adaptation, also in the Slovenian context, include the institutional and legal framework(s), organization of palliative care, and the level of public awareness of palliative care, as discussed above. In terms of palliative care organization, McGrath and Holewa [44], highlight the following important factors: equity (equal access); autonomy and empowerment (respect for the patient’s choice); trust (recognition of and respect for the historical context, and empathy during the provision of care); humane approach (non-judgmental care with an emphasis on quality of life and choice for patients and their families); high quality of care (involvement of a multidisciplinary team of healthcare professionals and community-based organizations working together throughout the care pathway); emphasis on living (rather than dying); and honoring of cultural identity (respect for cultural practices, beliefs, and lifestyle). Thus, even though some steps have been undertaken in this direction, including through the Last Aid course and the PO-LAST project, much still needs to be done in Slovenia. Therefore, a key future challenge for the implementation of the Last Aid course in Slovenia is to take greater account of the country’s socio-cultural diversity, which stems from the fact that the course has been developed in an international environment. Consequently, an important objective is to pay more attention to the needs of individual local communities. In the past, the importance of responding to local needs was highlighted by both international [43,43]. and Slovenian researchers [45]. The need for greater socio-cultural diversity in the delivery of the Last Aid course in Slovenia is also reflected in the fact that we were unsuccessful in attracting course participants with lower levels of education and representatives of minority ethnic groups (e.g., the Roma ethnic community).

### Limitations of the Study

Selection bias could be one limitation of the study since all informants chose to participate in our course voluntarily. The fact that informants were asked about their opinion directly after the course limits the recall bias, but it does not provide any information about the impact of the course on the provision of palliative care. We also did not reach people with lower education, possibly partly due to the pandemic, which led to altered formats and delivery modes of the course, and partly due to rather limited advertising opportunities in local communities. However, the data we were able to analyze are interesting not only because of the extremely positive evaluations of the course but also because of the suggestions for organizing future discussions and training on this topic.

## 7. Conclusions

The Slovenian Last Aid experience is comparable to that reported by countries where the course had been previously organized. In all those countries, the course has raised awareness among the lay public about death, dying, and palliative and end-of-life care. In Slovenia, the courses were extremely well-received and favorably rated, with participants lauding both content and delivery. However, the pandemic has impacted the in-person experience and encouraged new, web-based formats. We are aware that, in the future, it may be necessary to develop standards for online delivery of the course and to ensure that this is promoted in different communities and considers the socio-cultural diversity of the local environment. Socio-cultural diversity is a part of the Slovenian reality, and this has to be taken into account when preparing community education programs [46].

Nevertheless, the adaptation to Slovenian cultural requirements with information from the local communities was supported by materials prepared by the PO-LAST project, which entailed cooperation among Slovenian medical and healthcare professional, hospice volunteers, and university students. We are convinced that this work has also contributed to the excellent acceptance of the Last Aid course by the Slovenian community, but we are aware there is still a substantial amount of work to be done.

## Figures and Tables

**Table 1 healthcare-10-01154-t001:** Last Aid participants—sociodemographic data.

		Number	Share (%)
Sex	Men	37	15.2
	Women	206	84.8
Average age (standard deviation) (in years)		50.44 (16.122)	
Age bracket	Young (under 35)	48	19.9
	Middle-aged (35–64 years)	139	57.7
	Elderly (65 years and over)	54	22.4
Level of education	Primary	3	1.8
	Secondary	62	37.6
	Tertiary	100	60.6

**Table 2 healthcare-10-01154-t002:** Participant ratings for the course and individual modules.

Rating	The Entire Course (*n* = 243)	Module 1 (*n* = 250)	Module 2 (*n* = 249)	Module 3 (*n* = 248)	Module 4 (*n* = 248)
1 unsatisfactory	1 (0.4%)	1 (0.4%)	1 (0.4%)	1 (0.4%)	1 (0.4%)
2 satisfactory	0 (0.0%)	1 (0.4%)	0 (0.0%)	0 (0.0%)	0 (0.0%)
3 good	2 (0.8%)	4 (1.6%)	4 (1.6%)	5 (2.0%)	3 (1.2%)
4 very good	27 (11.1%)	38 (15.2%)	50 (20.1%)	33 (13.3%)	33 (13.3%)
5 excellent	213 (87.7%)	206 (82.4%)	194 (77.9%)	209 (84.3%)	211 (85.1%)
Average (SD)	4.86 (0.436)	4.79 (0.522)	4.75 (0.518)	4.81 (0.493)	4.83 (0.466)

SD—standard deviation

**Table 3 healthcare-10-01154-t003:** Participant ratings by age bracket.

	Young	Middle-Aged	Elderly	Kruskal-Wallis H Test
	Average (SD)	Average (SD)	Average (SD)	
The entire course	4.78 (0.471)	4.93 (0.264)	4.76 (0.687)	H = 6.360; df = 2; *p* = 0.042 *
Module 1	4.65 (0.565)	4.87 (0.398)	4.70 (0.735)	H = 9.993; df = 2; *p* = 0.007 *
Module 2	4.58 (0.613)	4.85 (0.354)	4.66 (0.717)	H = 1.836; df = 2; *p* = 0.004 *
Module 3	4.79 (0.504)	4.86 (0.387)	4.67 (0.718)	H = 4.071; df = 2; *p* = 0.131
Module 4	4.73 (0.536)	4.89 (0.314)	4.76 (0.687)	H = 4.437; df = 2; *p* = 0.109

* Indicates statistically significant values at *p* < 0.05.

**Table 4 healthcare-10-01154-t004:** Participant ratings (the entire course and individual modules) by sex (Mann–Whitney U test).

	Men	Women	Mann–Whitney U Test
	Average (SD)	Average (SD)	
The entire workshop	4.69 (0.796)	4.89 (0.332)	U = 3.093,5; *p* = 0.102
Module 1 Care at the End of Life	4.59 (0.798)	4.82 (0.456)	U = 3.176,0; *p* = 0.033 *
Module 2 Advance Care Planning and Decision-making	4.51(0.804)	4.80 (0.437)	U = 2.967,0; *p* = 0.007 *
Module 3 Symptom Management	4.69 (0.749)	4.82 (0.442)	U = 3.342,5; *p* = 0.284
Module 4 Cultural Aspects of Death and Bereavement	4.69 (0.749)	4.85 (0.398)	U = 3.283,0; *p* = 0.173

* Indicates statistically significant values at *p* < 0.05.

**Table 5 healthcare-10-01154-t005:** Thematic analysis of open-ended questions—results.

Theme	Categories	Code
Time	Period before death Period after death Perceived need Support for the bereaved	48
Suggestions	Ethical and cultural aspects Regular education Spirituality Children	35

## Data Availability

The data presented in this study are available on request from the corresponding author. The data are not publicly available due to (language of it).

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
