# Peer review of "Last Aid Course—The Slovenian Experience"

_healthcare, 2022, doi:10.3390/healthcare10071154_

Round 1

Reviewer 1 Report

The authors use the term palliative care and terminal care interchangeably in the manuscript. Terminal care, that is care provided to those in the final stages of their illness is but one part of palliative care which, ideally, has a role to play at the time of diagnosis of any life-limiting illness. People need not be actively dying to benefit from palliative care. A major source of confusion about this exists not only in the lay public but among health care providers as well, and to some extent is perpetuated in this manuscript.

The overall aim of the study is stated, but no specific research questions related to it are articulated.

More specific information needs to be provided about the survey that was conducted.

A pre-existing survey was used. Is there any information about its psychometric properties? 

  • How was the survey sent to applicants? Via email? Were reminder notices sent for non responders?
  • When was the survey sent to webinar participants? Immediately following the webinar?
  • How many surveys in total were sent out and how many responded? What was the response rate?
  • Do the researchers have a sense if non-responders might have differed in some significant way to those who did complete the survey?

The second section of the survey summarized personal reflections of the participants  and consisted of six sub-questions exploring their understanding of the specific topics  covered in the course, and suggestions for future work. How exactly was this qualitative data analyzed?  The manuscript requires more description about this. What steps were taken to ensure rigor as it is understood from a non-positivist perspective?

The authors claim in the abstract that they used a mixed methods approach. There is no evidence in the paper to support this. Collecting quantitative and qualitative data in the same study is not the same thing as conducting a mixed methods study. The authors need to:

  • Provide an adequate rationale for using a mixed methods design to address the research question.
  • Explain how the different components of the study are effectively integrated to answer the research question.
  • Clearly demonstrate and explain how qualitative and quantitative phases, results, and data were integrated.
  • Speak to the issue of meta-inference, that is, the overall interpretations derived from integrating qualitative and quantitative findings. Meta-inference occurs during the interpretation of the findings from the integration of the qualitative and quantitative components and shows the added value of conducting a mixed methods study rather than having two separate studies. 
  • Are there divergences and inconsistencies between quantitative and qualitative results adequately? If so, how do the authors explain these?
  • Do the different components of the study adhere to the quality criteria of each tradition of the methods involved? How?

Author Response

Dear reviewer,

We added the suggested change to the manuscript.

With kindly regards,

Erika Zelko

Reviewer 2 Report

I would sincerely thank the authors for the article which is clear and well written; moreover it adds evidences about public health approach in palliative care (Kellehear sensu). The aims of the paper are consistent with the results presented. I would also congratulate with the organizations and those within them contributing to implement Slovenian LAST AID courses: tackling with the “non-professionals” to raise the awareness of palliative care and to overcome the taboo of death is one of the corner stone of public health.

That said, I have identified some aspects that could be deepened or integrated to increase the quality of the article.

1.       P. 7 – lines 258-261:  I would suggest, if available, to provide a more detailed comparison between Slovenian data and other data collected through the survey in other country which have implemented Last Aid courses.

2.       To better situate the project and the results of the survey would be useful to know some more about Slovenian state of art in palliative care organizations, especially in those area project has been carried out. Since developing PC, death and dying awareness through compassionate cities or LAST AID projects should rest on a developed network of services, to tackle with the attended increase of service demands, I would suggest to better describe the context: for instance, how many patients access the PC services? Which type of services are available nowadays (Home care, Hospice, Hospital PC teams, Nursing homes)?

3.       It would be interesting a benchmark between the statistics on the sample population and those of the Slovenian population as a whole. As you hypothesized a connection between education level and involvement in the project, which it could be convincing, would you provide to the readers other useful information (i.e. Do they have assisted patients before joining the courses? Do they recently loss someone?) or associations between variables.

4.       It is not clear to me the characteristics of the sample and the qualitative methodologies: the 250 questionnaires collected refer to all the participants? How long after the courses participants filled the questionnaire? How many years questionnaires refer to? Could you add something more and explain how you managed qualitative analysis of the open-ended answers?

5.       Table 4: check typos on the use of “,” and “.”

Author Response

(The authors gave the same response as above.)

Reviewer 3 Report

This study was designed to analyze the participants’ responses in the Last Aid course in Slovenia. This course is designed to educate and raise awareness of palliative care among the lay members of the public.

The study focuses on a relevant subject, but some revisions are necessary:

-          The title is not indicative of the study

-          The purpose of the study is unclear

-          the methods aren't adequately described

o   who are the study participants? Does the study cover the population of Slovenia?

o   How was the sample obtained? Is it representative of the population?

o   When and how did the participants get access to the questionnaires?

o   How was the qualitative information analyzed and validated?

-          The results show significant data that are not discussed (table 3 and table 4).

-          I would like to see a table with the qualitative analyses.

-          The discussion and conclusion are very general and not well supported by the results of the study.

Author Response

(The authors gave the same response as above.)

Round 2

Reviewer 2 Report

Authors have answered to all the comments I made. Although there may be aspects that can be improved, they refer to the design of the study conducted and the choice of the tools adopted, therefore not attributable to the paper itself. However, the authors have made the limitations explicit, allowing the reader to consider potential areas for improvement.

I would only suggest to check on some typos consequent to corrections made.

As far as it concerns to me, once typos have been corrected, the paper can be published.

Author Response

I added our reply as file.

Reviewer 3 Report

The limitations of the study should be specified. The limitations described are directed to the course under evaluation and not to the study conducted.

Author Response

I added our reply as file.
